# Continuous Glucose Monitoring in Patients Following Simultaneous Pancreas–Kidney Transplantation: Time in Range and Glucose Variability

**DOI:** 10.3390/diagnostics13091606

**Published:** 2023-04-30

**Authors:** Ilya V. Dmitriev, Anastasia S. Severina, Nikita S. Zhuravel, Madina I. Yevloyeva, Rustam K. Salimkhanov, Svetlana P. Shchelykalina, Evgeniy A. Bezunov, Minara S. Shamkhalova, Julia F. Semenova, Vadim V. Klimontov, Marina V. Shestakova

**Affiliations:** 1Sklifosovsky Research Institute for Emergency Medicine, 129090 Moscow, Russia; ildmi@mail.ru; 2Endocrinology Research Center, 117292 Moscow, Russia; 3Department of Medical Cybernetics and Computer Science MBF Pirogov Russian National Research Medical University (RNRMU), 117997 Moscow, Russia; 4FSBI “Central Clinical Hospital with Polyclinic” of the Presidential Department of the Russian Federation, 121359 Moscow, Russia; 5Research Institute of Clinical and Experimental Lymphology—Branch of the Institute of Cytology and Genetics, Siberian Branch of Russian Academy of Sciences (RICEL—Branch of IC&G SB RAS), 630060 Novosibirsk, Russia

**Keywords:** pancreas transplantation, continuous glucose monitoring, type 1 diabetes mellitus

## Abstract

Simultaneous pancreas–kidney transplantation (SPKT) can improve long-term patient survival and restore endogenous insulin secretion in recipients with type 1 diabetes (T1D). There are currently few data on glucose fluctuations assessed by continuous glucose monitoring (CGM) after SPKT. Aim: to evaluate CGM-derived time in range (TIR) and glucose variability (GV) in patients with T1D and functioning pancreatic grafts after SPKT. Fifty-four CGM recordings from 43 patients, 15 men and 28 women, aged 34 (31; 39) years were analyzed. Time since SKPT was up to 1 year (group 1, *n* = 13), from 1 to 5 years (group 2, *n* = 15), and from 5 to 12 years (group 3, *n* = 26). TIR (3.9–10 mmol/L), Time Above Range (TAR), Time Below Range (TBR), and GV parameters were estimated. There were no differences in mean glucose (5.5 [5.1; 6.2], 5.9 [5.4; 6.2], and 5.9 [5.6; 6.7] mmol/L), TIR (97.6 [92.8–99.1], 97.2 [93.2; 99.1], and 97.5 [93.4; 99]%); TAR (0, 1.8 [1.3; 3.7], and 2.5 [2; 5]%), TBR (5 [3.3; 12.7], 4.1 [2.2; 10.1], and 3.5 [1.3; 6.5]%) and GV parameters between three groups (all *p* > 0.05). Thus, recipients with functioning pancreatic grafts demonstrate remarkably high TIR and low GV after SPKT.

## 1. Introduction

Pancreas transplantation is the only treatment option for type 1 diabetes (T1D) patients that provides optimal glycemic control and restores insulin independence. Successful pancreas transplantation eliminates the daily stigma and burden of diabetes and improves quality of life and life expectancy [1]. However, pancreatic graft dysfunction induced by rejection, toxicity of calcineurin inhibitors, and some other factors may cause a relapse of hyperglycemia. Therefore, an assessment of daily glucose fluctuations in the early and late post-transplantation period can be an effective tool to detect pancreatic graft dysfunction.

Continuous glucose monitoring (CGM) is increasingly being used in diabetes management. This method provides a comprehensive assessment of glucose fluctuations, including episodes of hypoglycemia and hyperglycemia. In 2002, Kessler L. et al. reported restoration of good glycemic control assessed by CGM in T1D subjects after simultaneous pancreas–kidney transplantation (SPKT) [2]. Since then, several studies with the use of CGM have been performed in pancreatic graft recipients. The results demonstrated the achievement of normoglycemia in the first days and weeks after transplantation [3,4,5]. At the same time, asymptomatic episodes of hypoglycemia were reported [3,6]. It was found that CGM-recorded hyperglycemia can be a sign and predictor of chronic pancreatic graft rejection [7]. Therefore, CGM may be a particularly useful tool in T1D patients after SPKT.

In recent years, standards have been developed for the digital evaluation of CGM data. As recommended by the International Consensus on the Use of CGM [8], percentages of Time in Ranges (TIRs; target, hypoglycemia, and hyperglycemia) and a measure of glucose variability (GV) should be reported as key diabetes control metrics in clinical studies. It has been shown that the Time in the target Range (TIR) increases up to 92% in the first 6 weeks after SPKT [5].

In recent years, the phenomenon of GV has attracted increasing attention. It has been shown that increased GV is associated with many biochemical and pathophysiological abnormalities involved in the pathogenesis of vascular complications and the phenomenon of metabolic memory in diabetes [9]. High GV was recognized as a risk factor for micro- and macrovascular diabetic complications and hypoglycemia [10,11,12]. Islet transplantation, as well as pancreatic transplantation, can reduce GV in patients with type 1 diabetes [2,5,13]. The GV phenomenon is characterized by the amplitude, frequency, and duration of the glucose fluctuations. Accordingly, a complete analysis of the GV requires a set of metrics that describe these characteristics [14,15]. To date, no study with in-depth TIR and GV analysis at various times after SPKT has been carried out.

Therefore, in this study, we aimed to assess CGM-derived TIRs and GV in T1D recipients with functioning pancreatic grafts at different times after SKPT.

## 2. Materials and Methods

Study design. We conducted a cross-sectional study. Patients were enrolled from March 2014 to October 2021 in two clinical centers: the Sklifosovsky Research Institute for Emergency Medicine and the National Medical Research Center for Endocrinology. The study cohort included patients who underwent SKPT in the period from 2008 to 2021 with pancreatic grafts with instant function at the time of the study. Deaths during hospitalization or in the remote postoperative period, loss of function of the pancreatic graft, and refusal or inability to participate in the study were applied as exclusion criteria.

From 1 January 2008 to 1 January 2021, the Sklifosovsky Research Institute for Emergency Medicine performed 79 SKPs in patients with T1D and end-stage renal disease.

The process of patient recruitment is shown in Figure 1.

Characteristics of recipients. Forty-three T1D patients, 15 men (34.9%) and 28 women (65.1%), aged 34 [31; 39] (median [25; 75 percentiles]) years, with body mass index (BMI) 20.8 [19.4; 23.5] m/kg^2^ were included. The mean daily insulin dose in our patients before transplantation was 40.6 [37.1; 46.2] IU. Before transplantation, the level of HbA1c was 7.5 [6.9; 8.2]%.

Characteristics of donors. We harvested organs during a multi-organ removal from donors, mostly men (*n* = 38, 88.4%), aged 28 [24; 34] years. All donors were brain dead: in 27 cases (62.8%) due to traumatic brain injury and in 16 (37.2%) cases due to stroke. The median timing was 7.5 [6; 10] hours for the preservation of kidney grafts (KG) and 9 [8; 10.5] hours for pancreatic graft (PG) preservation.

Immunosuppression. Patients received triple immunosuppressive therapy including calcineurin inhibitors (tacrolimus, cyclosporine), antimetabolites (mycophenolate mofetil, mycophenolic acid), and glucocorticoids (prednisolone). Tacrolimus was the most commonly used calcineurin inhibitor in basic IST (*n* = 42). As an induction IST, monoclonal antibodies (basiliximab) were used in 32 patients, and polyclonal antibodies were applied in 11 recipients (rabbit antithymocyte globulin or equine antithymocyte globulin).

Groups. We performed daily monitoring for various time periods post-transplantation, the median of which was 4.7 [1.1; 8.6] years. Based on the time since transplantation, all CGM recordings were divided into 3 groups: I—with study periods up to 1 year after transplantation (*n* = 13), II—from 1 year to 5 years (*n* = 15), and III—over 5 years (*n* = 26). Time since transplantation in these groups was 0.4 [0.2; 0.8], 3.3 [2.6; 3.8], and 8.8 [7; 9.9] years, respectively.

CGM. We performed blind CGM using the iPro2 monitoring system and CareLink^®^ iPro software (Medtronic, Minneapolis, MN, USA). Before the CGM started, we instructed the participants on the calibration rules and other aspects of the procedure. CGM ran in the background with no change to the usual diet, physical activity, and work schedule. At the time of the study, all patients were considered normoglycemic and were not receiving any antihyperglycemic therapy. The median number of CGM values was 821 [751; 865] and the median recording time was 68.4 [61; 72] h.

The following parameters were derived from the CGM data: mean glucose, TIR (3.9–10 mmol/L), Time in Tight range (TTR, 3.9–7.8 mmol/L), Time Above Range (TAR; 10–13.9 mmol/L, >13.9 mmol/L), Time Below Range (TBR, 3.0–3.8 mmol/L, <3.0 mmol/L), Standard Deviation (SD), Mean Amplitude of Glycemic Excursions (MAGE), 2-h Continuous Overlapping Net Glycemic Action (2-h CONGA), Lability Index (LI), J-index, Mean Absolute Glucose (MAG), M-value, High Blood Glucose Index (HBGI), and Low Blood Glucose Index (LBGI). The clinical significance of these indices has recently been reviewed [13,14,15]. Briefly, SD, CV, MAGE, LI, and MAG reflect the variability in general; CONGA, J-index, M-value, and HBGI are most closely associated with hyperglycemia, while LBGI indicates the risk of hypoglycemic events. The GV parameters were estimated with the EasyGV v. 9.0 calculator proposed by N. Hill et al. [16]. Evaluation of TIR, TAR, and TBR were performed according to the recommendations from the International Consensus on Time in Range [17]. All parameters were calculated for 24-h, daytime (6.00–23.59), and nocturnal (0.00–5.59) periods. Additionally, the number and duration of the episodes of hypoglycemia and hyperglycemia were estimated. We recognize hypoglycemia as a decrease in interstitial glucose levels < 3.9 mmol/L for at least 15 min [18].

Ethical aspects. The study was approved by the Local Ethics Committee of the Sklifosovsky Research Institute for Emergency Medicine (Protocol No. 3-13 of 22 July 2013). All participants provided their written informed consent prior to the study procedures.

Statistical analysis. The package STATISTICA 10.0 (StatSoft Inc., Tulsa, OK, USA, 2011, CSHA) was used for statistical analysis. We checked the normality of the distribution using the Shapiro–Wilk test. When comparing two independent groups on quantitative grounds, the Mann–Whitney criterion was applied; for two related groups, the Wilcoxon criterion was used; and for three independent groups, the Krasker–Wallis criteria were used. The differences in *p* < 0.05 in onetime comparisons and 0.017 in pairs, considering the Bonferroni correction, were considered statistically significant.

## 3. Results

Clinical characteristics of the study participants. Most patients had T1D from their childhood or adolescence, the mean age of diabetes onset was 11 [7; 14] years. The mean diabetes duration at the time of SPKT was 24 [20; 29] years. Three patients underwent pre-dialysis transplantation, the others were on renal replacement therapy for 2 [1; 3] years. Among them, 29 individuals were on hemodialysis and 11 were treated by permanent outpatient peritoneal dialysis. Sixteen (37.2%), 19 (44.2%), 7 (16.3%), and one (2.3%) patient had blood type O, A, B, and AB, respectively.

Groups of patients were comparable in all parameters considered (Table 1).

Parameters of CGM. 24-h, nocturnal, and daytime CGM parameters are presented in Table 2. The mean monitored 24-h glucose was 5.9 [5.4; 6.3] mmol/L and TIR (3.9–10 mmol/L) was remarkably high: 97.5 [93.4; 99]%.

The episodes of high interstitial glucose (>10 mmol/L) were revealed in 14 recordings (25.9%). The total duration of these episodes was 105 [55; 240] min. The duration of the longest episode was 60 [40; 160] min. Two episodes of hyperglycemia > 13.9 mmol/L, both lasting 15 min, were revealed in one subject.

The episodes of low glucose (<3.9 mmol/L) were revealed in 24 recordings (44.4%). The total duration of the episodes was 195 [95; 300] min, and the maximum duration was 95 [60; 175] min. The episodes of glucose levels < 3.0 mmol/L occurred in six cases (11%). The total duration of these episodes was 160 [65; 165] min, and the maximum duration was 150 [45; 160] min.

The CGM parameters in the groups of recipients with different times since SPKT are presented in Table 3.

The values of TIR (3.9–10 mmol/L) were 97.6 [92.8; 99.1], 97.2 [93.2; 99.1], and 97.5 [93.4; 99]% in groups I, II, and III, respectively (*p* for interaction = 0.74). TTR (3.9–7.8 mmol/L) was 92 [86; 97.1]% in group I, 93 [85.6; 96.8]% in group II, and 91 [79.8; 96.8]% in group III (*p* for interaction = 0.88).

The episodes of hyperglycemia (>10 mmol/L) occurred only in patients from groups II and III. In group II, episodes of hyperglycemia were recorded in five cases (31.3%). The total duration of all episodes and the maximum duration of the longest episode were 60 [60; 180] min and 60 [60; 160] min, respectively. In group III, episodes of hyperglycemia above >10 mmol/L occurred in ten cases. The total duration of all episodes and the maximum duration of the longest one were 117.5 [40; 240] min and 77.5 [40; 105] min, respectively. There were no significant differences between the groups in the number and total duration of the episodes of hyperglycemia (both *p* = 0.95) and the duration of the longest episode (*p* = 0.53). The values of TAR in groups II and III were 1.8 [1.3; 3.7] and 2.5 [2; 5]%, respectively. No differences between the groups were found (*p* for interaction = 0.79). One episode of hyperglycemia (>13.9 mmol/L) was revealed in a patient who underwent SPKT 7.4 years ago.

In group I, asymptomatic hypoglycemia (<3.9 mmol/L) occurred in seven recordings (58.3%). The median number of episodes was four [2; 6]. The total duration of the episodes and the maximum duration of the longest one were 215 [50; 315] and 85 [30; 140] min, respectively. Two episodes of glucose levels < 3 mmol/L occurred in two patients (8.3%). The duration of the episodes was 35 and 25 min.

In group II, episodes < 3.9 mmol/L occurred in six cases (37.5%). All episodes were asymptomatic, with a mean number of four [2; 6]. The total duration of the episodes and the maximum duration of the longest one were 185 [135; 300] min and 107.5 [65; 125] min, respectively. Two episodes of hypoglycemia < 3.0 mmol/L were recorded in two subjects. The duration of the longest episode was 45 min. Glucose levels recovered spontaneously in all cases.

In group III, episodes of low glucose < 3.9 mmol/L occurred in 12 cases (46.2%). The median number of episodes was 1.5 [1; 3.5]. The total duration of all episodes and the maximum duration of the longest episode were 172.5 [75; 310] min and 122.5 [52.5; 224.5] min, respectively. We revealed hypoglycemia < 3.0 mmol/L in two individuals. One patient had an episode lasting 160 min, another one had five episodes. The total duration of the episodes and the duration of the longest episode were 395 and 165 min, respectively. All episodes were asymptomatic and recovered spontaneously.

The value of TBR 3.0–3.8 mmol/L was 4.5 [3.3; 12.0], 3.1 [2.2; 3.7], and 2.95 [1.05; 5.45]% in groups I, II, and III, respectively. There were no statistically significant differences between the groups (*p* for interaction = 0.26). The value of TBR < 3 mmol/L in group I was 0.65 [0,6;0,7]%; in group II, it was 4,6 [3,4;5,7]%; and in group III, it was 7 [3.9%;10.1]%. There were no statistically significant differences (*p* for interaction = 0.11).

## 4. Discussion

In this study, we assessed glucose fluctuations with CGM in recipients with T1D with good pancreatic graft function. The results demonstrate remarkably high TIR and low GV at different times after SPKT.

Long-term survival of a kidney transplant is determined by a complex of factors, including the control of blood pressure, lipid spectrum, anemia, the duration of the dialysis period, the quality of the donor organ, the severity and quality of management of other complications of diabetes, the state of the cardiovascular system, and the efficacy of immunosuppression, with the leading role being glycemic control. Glucose homeostasis again comes to the fore, being temporarily pushed to the second position in the terminal period of diabetic nephropathy by the problems of RRT [19]. Awareness of this need to prevent the development of recurrent nephropathy and the progression of complications of diabetes has led to an increase in the popularity of SKPT, despite the remaining risks.

To date, there is a shortage of data on the assessment of glycemic control with CGM in subjects with T1D after SKPT. Rodríguez LM et al. reported that CGM parameters in T1D patients after SKPT are comparable with those of healthy subjects [4]. However, asymptomatic nocturnal hypoglycemia in recipients was also reported [6]. Interestingly, it was demonstrated that mean monitored glucose can predict pancreatic graft function in T1D patients treated with SPKT or pancreas transplantation alone [7]. In a study performed in the Mayo Clinic TIR (3.9–10 mmol/L), in 92% of T1D subjects treated with SPKT, 93.4% with pancreas after kidney transplantation, and 88.5% with pancreas transplantation alone, the values of TBR < 3.0 mmol/L were only 0.3%, 1.5%, and 0.3%, respectively [5].

Our study is not the first but the largest one in the field (43 patients, 54 CGM recordings) and includes a series of observations in different periods after SPKT, from one week to 12 years. The mean 24-h glucose level in our patients was 5.9 [5.4; 6.3] mmol/L, and the mean TIR (3.9–10 mmol/L) was 97.5 [93.4; 99]%. We also evaluated a panel of GV indices for an in-depth analysis of glucose fluctuations. The values of SD, MAGE, 2-h CONGA, MAG, LI, J-index, M-value, HBGI, and LBGI were low and comparable to those obtained in young and middle-aged Russian individuals with normal glucose tolerance [20].

Unlike other authors [3,5], we found very high TIR and low TAR and TBR values (medians, 97.5, 3.6, and 2.5%, respectively) in our subjects. Mean values of TIRs and other CGM parameters, including GV indices, did not differ in groups of recipients with different terms after transplantation. However, the episodes of hyperglycemia were revealed only in patients with a post-transplant period of more than a year. This may reflect the hyperglycemic effect of immunosuppressive therapy or subclinical decline of the pancreatic graft function. Recently, it was demonstrated that weight gain between 6 and 36 months after pancreas transplantation is directly associated with fasting glucose and HbA1c at 36 and 60 months [21]. However, in our cohort, BMI was similar in the groups of recipients with different times after transplantation. The change in pancreatic graft function over time and its effect on glucose fluctuations remains to be clarified. It was shown that 10 years after SPKT, only 56% of recipients demonstrate normoglycemia without insulin treatment [22]. In this regard, CGM may be a useful tool for detecting early signs of pancreatic graft dysfunction. Mittal et al. [3] showed that some recipients with early hyperglycemia have normal HbA1c after a year of follow-up, whereas patients with an increase in HbA1c after 1 year demonstrate early hyperglycemia with CGM.

Like Dadlani et al. [5], we recorded some episodes of low glucose in patients after successful SKPT. Hypoglycemic episodes were recorded in recipients at different times after transplantation without any differences in the number and total duration of the episodes between the groups. Episodes of glucose < 3.0 mmol/L were noted in two patients in each group. TBR < 3 mmol/l were 0.6% and 0.7% in patients from group I, 5.7% and 3.4% in patients from group II, and 3.9% and 10.1% in patients from group III. All recorded episodes were asymptomatic; glucose levels recovered spontaneously. Long diabetes duration is considered to be a risk factor for defective glucose counter-regulation in T1D subjects, with impaired glucagon secretion as an important mechanism. It was shown that transplantation of the whole pancreas can restore the glucagon response to hypoglycemia, whereas islet transplantation has produced conflicting results with potential partial recovery of glucagon responses to hypoglycemia. The reaction of glucagon after transplantation manipulations is most pronounced in the hierarchical activation of anti-regulatory hormones and symptoms [23]. However, in several studies, glucagon secretion after SKPT was not restored completely or was improved, but complete normalization was not observed [24,25].

The causes of hypoglycemia in organ recipients are a matter of debate. In these patients, a restriction of carbohydrate intake due to the fear of hyperglycemia, peripheral hyperinsulinemia, defective glucose counter-regulation, the production of insulin antibodies increasing the half-life of circulating insulin, and abnormal growth of pancreatic islet cells can contribute to hypoglycemic events. A change in the venous drainage of the pancreatic graft [6,26,27] plays an important role in the increase in peripheral hyperinsulinemia.

The cross-sectional design and short CGM duration are obvious limitations of our study. We did not evaluate the effect of diet, physical activity, or insulin secretion on CGM parameters. Changes in glycemic control in the early post-transplantation period were not taken into account. Future studies with longer follow-ups and duration of CGM are urgently needed.

## 5. Conclusions

In this study, we conducted an in-depth analysis of glucose fluctuations with CGM in patients with T1D at different times after SPKT. The results demonstrate that in recipients with a functioning pancreatic graft, high values of TIR and low GV are maintained even in the long-term period. Only a small percentage of patients have episodes of high or low glucose levels. The data provide further support for the notion that assessment of glucose fluctuations with CGM can be a useful tool to monitor pancreatic graft function.

## Figures and Tables

**Figure 1 diagnostics-13-01606-f001:**
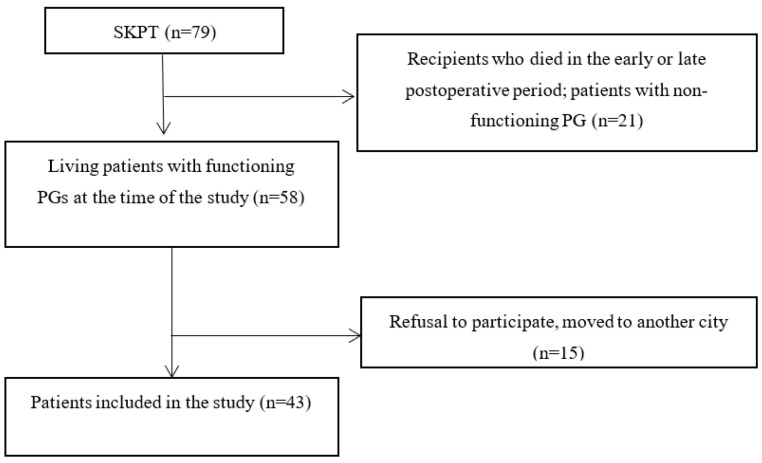
Flow chart of the study patient recruitment. SKPT, simultaneous kidney–pancreas transplantation; PG, pancreatic graft.

**Table 1 diagnostics-13-01606-t001:** Clinical characteristics of the study groups.

Characteristics	Group I (*n* = 13)	Group II (*n* = 15)	Group III (*n* = 26)	*p* *
Age, years	35.5 [34; 37]	33.5 [31; 36]	34 [30; 39]	0.53 **
Sex, m/f, n (%)	3 (25)/9 (75)	4 (25)/12 (75)	9 (34.6)/17 (65,4)	0.74 ***
BMI, kg/m^2^	21.1 [18.7; 22.8]	20.9 [19.6; 24.1]	20.5 [19.4; 22]	0.74 **
Diabetes duration, years	26.5 [20.5; 30.5]	21.5 [20; 25]	25 [20; 32]	0.11 **
Renal replacement therapy before SPKT				
CAPD, n (%)	5 (41.7)	3 (18.8)	5 (19.2)	0.27 ***
HD, n (%)	6 (50)	13 (81.2)	16 (61.5)	0.20 ***
CAPD + HD, n (%)	0 (0)	0 (0)	3 (11.5)	0.51 ***
No dialysis, n (%)	1 (8.3)	0 (0)	2 (7.7)	0.18 ***
RRT duration, years	2 [1; 4.3]	2 [2; 3]	2 [1; 3]	0.57 **
Blood group				
O(I)	5 (41.7)	6 (37.5)	12 (46.2)	0.86 ***
A(II)	6 (50)	6 (37.5)	8 (30.8)	0.52 ***
B(III)	1 (8.3)	2 (12.5)	5 (19.2)	0.65 ***
AB(IV)	0 (0)	2 (12.5)	1 (3.8)	0.31 ***
Donor age, years	32 [27; 35]	27 [24; 29]	28.5 [24; 33]	0.20 **
Sex of the donor, m/f, n (%)	10 (83.3)/2 (16.7)	15 (93.8)/1(6.2)	23 (88.5)/3 (11.5)	0.68 ***
Stroke/TBI	6 (50)/6 (50)	8 (50)/8 (50)	6 (23.1)/20 (76.9)	0.12 ***
Intra-abdominal/retroperitoneal localization of PG	0 (0)/12 (100)	1 (6.3)/15 (19.7)	5 (19.2)/21 (80.8)	0.16 ***
Y-graft/ISABS	11 (91.7)/1 (8.3)	13 (81.3)/3 (18.7)	25 (96.2)/1(3.8)	0.26 ***
Portal/caval venous outflow of PG	0 (0)/12 (100)	3 (18.8)/13 (81.2)	3 (11.5)/23(88.5)	0.29 ***
Duodenojejunal/duodenoduodenoanastomosis	4 (33.3)/8 (66.7)	2 (12.5)/14 (87.5)	6 (23.1)/20 (76.9)	0.42 ***
KG CIT	8 [6.8; 10.5]	8 [6.3; 12.3]	7.5 [5.5; 9.5]	0.45 **
PG CIT	9.1 [8; 9.8]	9 [7.8; 11]	9.3 [8; 11]	0.52 **
Tacrolimus	12(100)	16 (100)	25 (96,2)	0.33 ***
Monoclonal antibodies	10 (83.3)	8 (50)	22 (84.6)	0.03 ***
Polyclonal antibodies	2 (16.7)	8 (50)	4 (15.4)	0.03 ***

* Me [25%, 75%], ** Kruskal-Wallis criterion; *** Pearson’s chi-squared criterion; RRT, renal replacement therapy; CAPD, continuous ambulatory peritoneal dialysis; HD, hemodialysis; TBI, traumatic brain injury; KG, kidney graft; CIT, cold ischemia time.

**Table 2 diagnostics-13-01606-t002:** 24-h, nocturnal, and daytime CGM parameters.

Parameter	24-h Parameters	Nocturnal Hours (0.00–5.59)	Daytime Hours (6.00–23.59)
Mean glucose, mmol/L	5.9 [5.4; 6.3]	5.3 [5; 5.9]	6.0 [5.6; 6.5]
TTR (3.9–7.8 mmol/L),%	91.5 [84.5; 96.8]	97.2 [86.1; 97.2]	92 [84; 97.4]
TIR (3.9–10 mmol/L),%	97.5 [93.4; 99]	97.2 [93.1; 97.2]	97.8 [93.7; 98.7]
TAR (10–13.9 mmol/L),%	2.45 [1.3; 4]	3.2 *	3.2 [1.7; 5.5]
TAR (>13.9 mmol/L),%	0.5 *	0	0.6 *
TBR (3–3.8 mmol/L),%	3.4 [2; 5.7]	8.9 [2.8; 12.5]	1.8 [0.9; 3.6]
TBR (<3 mmol/L),%	3.4 [0.7; 5.7]	14.4 [2.8; 21.5]	2 [1.2; 3.1]
SD, mmol/L	0.9 [0.7; 1.2]	0.6 [0.5; 0.8]	1 [0.7; 1.2]
MAGE, mmol/L	16 [12.9; 18.6]	1.4 [0.8; 1.9]	2.2 [1.6; 2.8]
CONGA, mmol/L	2.1 [1.7; 2.8]	4.7 [4.2; 5.4]	5.2 [4.9; 5.7]
MAG, mmol × L^−1^ × h^−1^	5.1 [4.8; 5.6]	0.8 [0.5; 1.2]	1.1 [0.9; 1.3]
LI, (mmol/L)^2^/h	0.95 [0.7; 1.2]	0.2 [0.1; 0.5]	0.6 [0.3; 0.9]
J-index, (mmol/L)^2^	0.5 [0.2; 0.8]	11.2 [9.5; 13.4]	15.8 [13; 19.4]
M-value	14.8 [12.3; 18.4]	1.7 [0.7; 5.1]	1.4 [0.6; 2.2]
HBGI	1.5 [0.8; 2.9]	0.03 [0; 0.4]	0.9 [0.3; 1.4]
LBGI	0.8 [0.3; 1.2]	1.6 [0.7; 3.4]	1.3 [0.6; 1.9]

TTR, Time in Tight Range; TIR, Time in Range; TAR, Time Above Range; TBR, Time Below Range; SD, Standard Deviation; MAGE, Mean Amplitude of Glycemic Excursions; CONGA-2, 2-h Continuous Overlapping Net Glycemic Action; MAG, Mean Absolute Glucose; LI, Lability Index; HGBI, High Blood Glucose Index; LBGI, Low Blood Glucose Index. The data are shown as median [25% percentile; 75% percentile]; *—an episode of glycemia was noted in one patient.

**Table 3 diagnostics-13-01606-t003:** CGM parameters in recipients depending on the time since SPKT.

Parameter	Group I (*n* = 13)	Group II (*n* = 15)	Group III (*n* = 26)	*p* **
Mean glucose, mmol/L	5.5 [5.1; 6.2]	5.9 [5.4; 6.2]	5.9 [5.6; 6.7]	0.25
TTR (3.9–7.8 mmol/L),%	92 [86; 97.1]	93 [85.6; 96.8]	91 [79.8; 96.8]	0.88
TIR (3.9–10 mmol/l),%	97.6 [92.8; 99.1]	97.2 [93.2; 99.1]	97.5 [93.4; 99]	0.74
TAR (10–13.9 mmol/L),%	0.0	1.8 [1.3; 3.7]	2.5 [2; 4]	
TAR (>13.9 mmol/L) *,%	0.0	0.0	0.5	N/A
TBR (3–3.8 mmol/L),%	4.5 [3.3; 12]	3.1 [2.2; 3.7]	3 [1.1;5.5]	0.50
TBR (<3 mmol/L),%	0.7 [0.6; 0.7]	3.4 [1.7; 5.7]	7 [3.9; 10.1]	0.10
SD, mmol/L	0.8 [0.7; 1.2]	1 [0.7; 1.1]	0.9 [0.8; 1.3]	0.59
MAGE, mmol/l	2 [1.8; 2.8]	2.3 [1.4; 2.5]	2.2 [1.7; 3]	0.79
CONGA, mmol/L	4.8 [4.4; 5.4]	5.2 [4.9; 5.5]	5.1 [4.9; 5.7]	0.17
MAG, mmol/l/h	0.9 [0.7; 1.1]	1 [0.7; 1.3]	1 [0.7; 1.2]	0.67
LI, (mmol/L)^2^/h	0.4 [0.2; 0.7]	0.5 [0.2; 0.6]	0.5 [0.3; 1]	0.53
J-index, (mmol/L)^2^	14.2 [11.1; 17.1]	14.3 [12.4; 17.8]	15.6 [13.6; 20]	0.25
M-value	2.7 [1.5; 3.8]	1.3 [0.5; 3.1]	1.4 [0.8; 2.2]	0.25
HBGI	0.6 [0.3; 0.9]	0.8 [0.2; 1.2]	0.9 [0.3; 1.7]	0.52
LBGI	2.2 [1.5; 3.3]	1 [0.6; 2.1]	1.2 [0.8; 1.9]	0.07

TIR, Time in Range; TAR, Time Above Range; TBR, Time Below Range; SD, Standard Deviation; MAGE, Mean Amplitude of Glycemic Excursions; CONGA-2, 2-h Continuous Overlapping Net Glycemic Action; MAG, Mean Absolute Glucose; LI, Lability Index; HGBI, High Blood Glucose Index; LBGI, Low Blood Glucose Index. The data are shown as median [25% percentile; 75% percentile]. * The episodes of glucose > 13.9 mmol/L were recorded in one patient only. ** Kruskal–Wallis criterion.

## Data Availability

Not applictable.

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
