# Peer review of "Continuous Glucose Monitoring in Patients Following Simultaneous Pancreas–Kidney Transplantation: Time in Range and Glucose Variability"

_diagnostics, 2023, doi:10.3390/diagnostics13091606_

Round 1

Reviewer 1 Report

I was happy to read the manuscript "Continuous Glucose Monitoring in Patients Following Simultaneous Pancreas-Kidney Transplantation: Time in Range and  Glucose Variability by  Dmitriev I.V. et al. The chosen topis is important and SPKT patients are important to be followed because they have a huge chance to life.

I saw that a lot of patients from the initial 79 group have died. Maybe a few words about the failure of SPKT could be discussed into the introduction section.

 I find the manuscript interesting and I believe it should be published after some improvements:

In Introduction section:

line 35, please explain acronims when they are first introduced: CNI for example

I think more should be discussed into the introduction about SPKT, failures and success, also maybe  more about GV.

Materials and methods.

I consider study design well chosen, but in some cases, so many acronims can became confusing, I suggest mentioning more their meaning.

In result section.

In Table 1, there are a lot of acronims that must be explain at least when first added.

There are too many tables, maybe some figures would be more easily to be followed. Table 7 could be presented as a figure.

In the Discussion section more articles should be cited an d more information should be displayed about CGM in various types of transplantations, and at various stages.

Again, acronims. Line 244, RRT, renal replacement therapy, mentioned in table 1, but not explained.

This is not the first study of the kind, so many others should be mentioned.

The diabetes period before transplantation may be discussed.

A conclusion must be placed in the end.

Reviewer 2 Report

In this paper Dmitriev et al. present results from continuous glucose monitoring (CGM) in SPKT patients. They evaluated time in range (TIR), -above range, -below range and glucose variability (GV) in 43 patients in three groups. The groups are characterized by the time after transplantation (1 year, 1-5 years and 5-12 years). The authors found no difference in the investigated parameters between all groups and conclude that SPKT patients demonstrate a high TIR and low GV.

The study and statistics are well done, however a healthy control group is missing. Also a group kidney transplant patients would have been interesting to compare. Nonetheless the authors provide relevant data on blood glucose levels in SPKT patients. The paper is well written and organized. I recommend to publish the study after minor revision.

Minor issues: Wording line 287/288 needs to be corrected.

Reviewer 3 Report

The paper is valuable in reporting the results of blood glucose measurements by CGM in diabetic patients after Simultaneous Pancreas-Kidney Transplantation, but there is a serious lack of information, mainly with regard to patient characteristics. The following comments require significant current revisions.

-Patient characteristics in Table 1 are seriously underinformed. What is the degree of diabetes status? What diabetes treatment is the patient receiving, HbA1c? What is their glycoalbumin? In particular, whether insulin is used as treatment, its dosage, and the timing of injections must be noted as they contribute significantly to the incidence of hypoglycemia at night.

-What was the setting of the diet when you performed the measurements in the CGM?

-Unify the number of decimal places in the P value.

Round 2

Reviewer 1 Report

I agree and I am ok with the new version of the manuscript!

Reviewer 3 Report

No further comment.